# Effect of Sintering Mechanism towards Crystallization of Geopolymer Ceramic—A Review

**DOI:** 10.3390/ma16114103

**Published:** 2023-05-31

**Authors:** Nur Bahijah Mustapa, Romisuhani Ahmad, Wan Mastura Wan Ibrahim, Mohd Mustafa Al Bakri Abdullah, Nuttawit Wattanasakulpong, Ovidiu Nemeș, Andrei Victor Sandu, Petrica Vizureanu, Ioan Gabriel Sandu, Christina W. Kartikowati, Puput Risdanareni

**Affiliations:** 1Faculty of Mechanical Engineering & Technology, Universiti Malaysia Perlis (UniMAP), Arau 01000, Malaysia; bahijahmustapa@gmail.com (N.B.M.); wanmastura@unimap.edu.my (W.M.W.I.); 2Centre of Excellence Geopolymer and Green Technology (CEGeoGTech), Universiti Malaysia Perlis (UniMAP), Kangar 01000, Malaysia; 3School of Engineering and Technology, Walailak University, Thasala, Nakhon Si Thammarat 80160, Thailand; nuttawit.wa@wu.ac.th; 4Department of Environmental Engineering and Sustainable Development Entrepreneurship, Faculty of Materials and Environmental Engineering, Technical University of Cluj-Napoca, B-dul Muncii 103-105, 400641 Cluj-Napoca, Romania; 5Faculty of Materials Science and Engineering, Gheorghe Asachi Technical University of Iasi, Blvd. D. Mangeron 71, 700050 Iasi, Romania; sav@tuiasi.ro (A.V.S.); peviz@tuiasi.ro (P.V.); ioan-gabriel.sandu@academic.tuiasi.ro (I.G.S.); 6Romanian Inventors Forum, Str. Sf. P. Movila 3, 700089 Iasi, Romania; 7Technical Sciences Academy of Romania, Dacia Blvd 26, 030167 Bucharest, Romania; 8Department of Chemical Engineering, Universitas Brawijaya, Malang 65145, Indonesia; christinawahyu@ub.ac.id; 9Department of Civil Engineering, Faculty of Engineering, Universitas Negeri Malang, Malang 65145, Indonesia; puput.risdanareni.ft@um.ac.id

**Keywords:** geopolymers, ceramics, crystallization, sintering mechanism

## Abstract

Globally, there is an increasing need for ceramic materials that have a variety of applications in the environment, for precision tools, and for the biomedical, electronics, and environmental industries. However, in order to obtain remarkable mechanical qualities, ceramics have to be manufactured at a high temperature of up to 1600 °C over a long heating period. Furthermore, the conventional approach presents issues with agglomeration, irregular grain growth, and furnace pollution. Many researchers have developed an interest in using geopolymer to produce ceramic materials, focusing on improving the performances of geopolymer ceramics. In addition to helping to lower the sintering temperature, it also improves the strength and other properties of the ceramics. Geopolymer is a product of polymerization involving aluminosilicate sources such as fly ash, metakaolin, kaolin, and slag through activation using an alkaline solution. The sources of the raw materials, the ratio of the alkaline solution, the sintering time, the calcining temperature, the mixing time, and the curing time may have significant impacts on the qualities. Therefore, this review aims to study the effects of sintering mechanisms on the crystallization of geopolymer ceramics, concerning the strength achieved. A future research opportunity is also presented in this review.

## 1. Introduction

Ceramics are typically made up of inorganic and non-metallic solids, like clay, and have been used as a raw material for products for more than a thousand years. They are well-recognized for having high manufacturing temperature requirements and relatively high melting temperatures. Ceramics are typically made from a variety of different elements such as O, B, C, and N or a mix of metallic and non-metallic elements, in the form of oxides, carbides, borides, and silicates. Because of their thermal insulation, lightness, huge specific surface area, and resilience to thermal stress, ceramics are unique among other materials in several exciting ways [1]. Ceramics are typically oxide compositions that are developed in both amorphous and crystalline phases. They exhibit porosity in the micro and larger size ranges, and many of their shapes are not completely dense. Clay gives conventional ceramics their hardness and ductility, silica determines their remarkable stability at high temperatures and higher melting points, and feldspar produces the glass phase after initial firing. Due to the numerous drawbacks of the conventional methods for producing ceramics, which include extremely high temperatures that can exceed 1600 °C [2], and the fact that they are not fully dense, but exhibit porosity at the micro and larger size range, a new method of producing ceramics using geopolymer has been introduced, as geopolymer technology possesses good fire resistance, chemical corrosion resistance, high mechanical strength, and excellent durability [3].

Joseph Davidovits used the term “geopolymer” in the 1970s to characterize the substance produced when aluminosilicate minerals polycondense in an alkaline activator [4,5]. Essentially, a silicate monomer repeating unit makes up a geopolymer (Si–O–Al–O). The primary requirement for developing a stable geopolymer is having raw materials that are highly amorphous, have an appropriate proportion of reactive glassy content, require little water, and release aluminum easily [6]. The silicon and aluminum in the aluminosilicate sources and setting additives must be activated by strong alkalis to partially or completely change the glassy structure into an extremely compact composite [7]. Common activators include water glass, NaOH, Na_2_SO_4_, K_2_CO_3_, KOH, and K_2_SO_4_, as well as a small quantity of cement clinker. Since more than a century ago, sodium silicate has been utilized for making industrial goods including catalysts, coatings, molded items, and special cement [8,9,10]. Fly ash, cement, lime, slag, or other multivalent metal ions that encourage silicate gelation and precipitation are mixed with the soluble silicate. The release of silicate and aluminate monomers increases with the amount of NaOH that comes into contact with the reactive solid substance [11,12,13]. Generally, materials mainly containing silica (SiO_2_) and alumina (Al_2_O_3_) are possible sources for geopolymer synthesis [14]. A synthetic alkali aluminosilicate compound with good binding properties was produced through the interaction of aluminosilicate compounds and an alkaline solution [15]. The chemical composition, microstructure, and phase content of an aluminosilicate source affects its activity. In addition, several factors, such as the type of application, cost, and availability of materials, must be considered when selecting the precursor to synthesize the geopolymer.

The polymerization process produces an amorphous inorganic material consisting of Si–O–Al bonds [16]. A rapid silica–alumina reaction that takes place in an alkaline environment is necessary for the polymerization in order to produce a three-dimensional polymeric chain. Studies show that the efficiency of producing geopolymer concrete is highly dependent on the activators, such as their alkali concentration and solution ratio, curing temperature, time, pH, liquid-to-solid ratio, and aluminosilicates resources [17]. One of the unique characteristics of geopolymer is that it offers an alternative way to prepare ceramics. Geopolymers can be transformed into ceramic materials by suitable heat treatment with a well-defined phase composition and proper properties. This method is preferable, as it requires lower heat treatment and provides better properties. Because of this, they can be utilized as precursors for high-temperature-resistant products, matrices for composite materials, technical ceramics, ceramic tiles, and construction ceramics [18]. 

Due to a novel method of synthesizing ceramics with the addition of geopolymer precursors that benefits compressive strength and provides low permeability, superior chemical resistance, and outstanding fire resistance behavior, the production of ceramics employing geopolymer materials is currently on the rise [19]. Most importantly, geopolymer can be advantageously used to produce geopolymer ceramics. Sintering causes the formation of ceramic products due to the formation of crystalline phases. Direct high-temperature sintering of cured monolithic geopolymers leads to significant shrinkage and cracking, reducing the final products’ strength [20]. Much research is conducted mainly to study the optimum temperature for the sintering process. However, this review focused on the effect of the sintering mechanism on the crystallization of geopolymer ceramic during the sintering process. Therefore, the behavior and properties of geopolymer are further discussed in this paper, followed by discussion of the previous research on sintering’s effects on the crystallization and properties of the geopolymer. Geopolymer ceramics are expected to show a promising performance when sintering occurs under optimum conditions.

## 2. Geopolymer Overview

Geopolymers have received considerable attention because of their low cost, excellent mechanical and physical properties, low energy consumption, and reduced greenhouse emissions during the geopolymerization process [21]. The method of producing geopolymers involves activating aluminosilicates by dissolving aluminosilicate precursors, hydrolyzing Al^3+^ and Si^4+^, and polycondensing [SiO_4_] and [AlO_4_]^−^ tetrahedra in alkaline, alkaline silicate, or alkaline carbonate solutions at temperatures close to room temperature (20–90 °C). This material, first introduced by Davidovits in the 1980s, provides greater mechanical strength, higher corrosion and fire resistance, and a lower environmental effect when compared to Ordinary Portland Cement (OPC) [22]. Geopolymers, sometimes referred to as alkali-activated binders, have been explored as a potential OPC replacement with little effect shown on the environment.

Understanding the process of geopolymerization can, therefore, be the key to controlling several factors, such as the degree of polymerization, porosity, and the mechanical properties of geopolymers, allowing them to be customized for particular uses. Geopolymer materials are produced by mixing waste products or naturally occurring aluminosilicate sources, such as metakaolin, fly ash, and volcanic ashes, in alkali-activated solutions, such as sodium hydroxide and sodium silicate. Highly alkaline conditions cause reactive aluminosilicates to dissolve rapidly, liberating free [SiO_4_]^−^ and [AlO_4_]^−^ tetrahedral units into the solution. Sharing oxygen atoms between tetrahedral units and polymeric precursors results in the formation of polymeric Si–O–Al–O linkages [23]. As a result, binding gels with high early strength, high temperature resistance, and a potentially low CO_2_ footprint have been produced [24]. The general geopolymerization reaction process in the presence of KOH/NaOH is:(1)$${{(Si}_{2}O_{5}{Al}_{2}O_{2})}_{n}+{nH}_{2}O+{OH}^{-}\to nSi{\left(OH\right)}_{4}+nAl{\left(\mathrm{OH}\right)}_{4}^{-}$$

This process releases water that is usually consumed during dissolution. When the geopolymer reacts, water is released, giving the mixture workability for handling. An alkali-activated solution attacks aluminosilicate materials, causing them to dissolve and break down; encourages species diffusion and monomer formation (coagulation/gelation); and, finally, polymerization and the stability of polymeric structures causes the material to harden. These three stages of the geopolymerization process can be separated into separate but overlapping phases. The initial reorganization, dissolution/oligomerization, incubation, percolation/hardening, and consolidation stages of the geopolymerization process were categorized into five stages by Rouyer et al. [25]. Steins et al. demonstrated that for geopolymer cement pastes prepared from metakaolin, a tiny amount of alkali activator hastens the breakdown of the material and the development of a stiff percolating network. A higher atomic number of alkali activators hastens the production of oligomers with stronger connections [26].

A type of cementitious material known as geopolymer paste was created by blending aluminosilicate components with an alkaline or alkaline silicate solution. Due to the ongoing search for high-performance and/or environmentally friendly substitutes for conventional Portland cement, geopolymer cement pastes have come under increasing scrutiny. Additionally, geopolymer cement pastes have shown exceptional qualities, such as high compressive strength, minimal shrinkage, speedy or gradual setting, resistance to chemical attacks, fire resistance, and low thermal conductivity. Additionally, aluminosilicate minerals were also frequently dissolved and polycondensed in alkali-activated solutions at room temperature or above to create the geopolymer. As a result, [AlO_4_]^−^ and [SiO_4_] tetrahedrons linked by oxygen bridges formed an amorphous 3D network in the geopolymer cement pastes. Previous research had demonstrated countercations’ involvement in a number of reactions that took place during the alkali-activated reaction or geopolymerization reaction, including the acceleration of aluminosilicate dissolution, stabilization of solution species and colloids, a reduction in electrostatic repulsion between the anions, and promotion of gel formation and rearrangement. Therefore, the initial reaction mixtures’ composition, including the aluminosilicate sources used, the type of alkali-activated solution used, and the preparation procedures (reacting/curing regime) greatly influenced the geopolymerization process [27].

### 2.1. Source Materials

Geopolymers are a class of materials synthesized by alkaline activation of an aluminosilicate source at an ambient or higher temperature. Geopolymers have the potential to be used in waste management, fireproofing, construction, military engineering, and even as biomaterials due to their excellent heavy metal immobilization, high-temperature stability, quick solidification with high strength, and biological compatibility. For the application of geopolymeric materials, knowledge of the geopolymerization process and its affecting elements is necessary. However, the exact process is not fully understood so far, although the involved mechanism has been studied in the last few decades [28].

The application of geopolymer materials as construction components has risen significantly in recent years, as this class of materials offers many advantages over other conventionally used materials. These materials have the potential to replace Portland cement in cementitious materials, one of the materials that emits the most CO_2_ globally, as well as natural raw materials, like clay, in ceramic materials. In addition, they provide mechanical properties obtained by activated alkali compounds, such as high mechanical strength and durability. As a result, using this category of materials offers both technological and environmental benefits.

The term “geopolymer materials” refers to substances made of silicon and calcium oxide that, when exposed to a strong alkaline solution with a pH above 14, undergo a chemical reaction and produce mechanical resistance. Alkali-activated materials and geopolymer materials are frequently grouped together, although it should be highlighted that, whereas geopolymers are built of aluminum silicates, alkali-activated materials are chemically based on calcium and silicon. Because of this, even with a similar chemical principle, alkali-activated materials and geopolymers should not be used interchangeably, as the chemical structures formed are very different [29]. Figure 1 below shows a division of the geopolymerization process. 

The aluminosilicate supplies used, the type of alkali-activated solution, and the preparation circumstances (reacting/curing regime) each had a significant impact on the geopolymerization process [27].

#### 2.1.1. Precursor

Many of the waste materials from various industrial and agricultural activities are used as precursors in geopolymer concrete. It is possible to argue that geopolymer concrete is eco-friendlier and more efficient at handling enormous amounts of waste produced by other sectors. The sustainability of geopolymers can be increased by utilizing locally accessible materials as precursors, such as laterite soil. As a result, employing geopolymers as a greener substitute for Portland cement composites would greatly cut down on raw material usage, greenhouse gas emissions, and waste management needs. The advantages of utilizing geopolymer concrete for various construction applications are presented in Figure 2.

Concrete is the most-used human-made material in the world. OPC is typically used as a base material prior to making concrete. Production of Portland cement is one of the main contributors to greenhouse gas emissions, thus contributing to air pollution. Consequently, there has been continuous research for alternative structures and building materials with lower carbon footprints [30]. 

In the meantime, a waste by-product produced when pulverized coal is burned in electric thermal power plants is currently widely accessible around the world. One billion tons of fly ash are reportedly created each year, contributing to anthropogenic pollution. Fly ash is a silica- and alumina-rich material that can be activated with an alkaline solution to create an aluminosilicate gel that serves as a binder in geopolymer concrete. Almost-spherical fly ash particles may readily flow and merge in combinations [31].

Rice husk ash (RHA) is an agricultural by-product obtained from the process of burning rice hulls [32], and is prominently used as a fuel for electricity generation. RHA is extremely rich in silica, which exists mostly in amorphous and to some extent in crystalline phases and is further affected by burning temperatures and duration. RHA is amorphous; however, the phases of silica vary depending on the burning temperature and ash generation technique.

One of the most widespread source materials containing aluminosilicate in geopolymer concrete (GPC) is metakaolin (MK) [33]. MK is formed by calcining natural clays (kaolin) at a moderate temperature. Through the process of heating, the relatively unreactive kaolin undergoes a transformation into highly reactive metakaolin. This alteration involves a modification of the crystalline structure, resulting in a disrupted, layered structure [34]. The layers of kaolinite are delaminated, resulting in the exposure of reactive sites on the surfaces of the particles.

In most cases, sodium hydroxide (NaOH) and sodium silicate, or potassium hydroxide (KOH) and potassium silicate (K_2_SiO_3_), are used as alkali activators in geopolymer mixes. Kaolinitic clay is typically calcined to produce metakaolin; however, the thermal cycle of calcination should ensure the optimal conversion of kaolin to metakaolin. The outcomes of the various tests under consideration were completely consistent, demonstrating that calcination increased pozzolanic reactions; further, the heat cycle of 800 °C for 5 h allowed for the production of the highest pozzolanic values. That, along with other merits, indicates that it should be valued as an ecologically friendly cement production technique [35,36,37]. 

Past research used proven raw materials, chosen to affect the properties of the produced geopolymer, as summarized in Table 1. Research from Bong et al. [38] reported the effects of using wollastonite instead of sand or as a geopolymer precursor replacement on the properties of a one-part (just-add-water) geopolymer mortar. In their study, wollastonite was used as the precursor to replace the original function of sand. The findings indicate that the flexural strength of the mixtures was greatly increased by substituting wollastonite for sand. However, depending on the amount of wollastonite used, the compressive strength either increased or remained constant. Nevertheless, the workability was improved and the setting time of the mixtures was significantly lowered when wollastonite was used as a replacement for the geopolymer precursor.

Furthermore, Ren et al. [39] investigated the durability-related performances of geopolymer composite materials that were synthesized by the alkali activation of metakaolin (MK) with reinforcement using wollastonite (WS), tremolite (TR), and short basalt (SBF). The results of this study point to the practicality of strengthening geopolymers by adding fibers and mineral particles as reinforcement. Additionally, this improved the resistance to sulphate and chloride attack. This finding was supported by Archez et al. [40], where it is reported that if wollastonite was used as a reinforcement in the geopolymer system, it induced a different polycondensation composition. By guaranteeing that the metakaolin dissolved more thoroughly, the wollastonite increased viscosity and improved mechanical characteristics. Contrarily, the glass fibers served as an anchoring site during the geopolymerization process, which caused the material to fail in regard to ductility. The presence of aluminum also made it possible to monitor the reactive mixture’s viscosity and setting time, and it had a substantial impact on the microstructure and compressive strength. As a result, the properties of the fresh and hardened geopolymer composites could be controlled by the initial formulation [40]. A clear vision of precursor sources can be seen in Figure 3 below.

#### 2.1.2. Alkali Activator

In addition to the precursor, the properties of the geopolymer structure are also influenced by the alkali activator. The type of ion involved in the activation reaction plays a crucial role in the development of the microstructure of the resulting geopolymer. 

For geopolymer cement pastes made from metakaolin, Steins et al. showed that a small alkali activator accelerated the dissolution of the metakaolin and the formation of a rigid percolating network, and that the resulting oligomers had stronger interactions when the alkali activator had a higher atomic number [27]. The preparation and control of the geopolymer’s properties were greatly influenced by the type, content, reactivity, and quantity of the raw materials [42].

Yuan et al. [27] studied the effects of alkali-activated ions on the geopolymerization process of geopolymer cement pastes. It was observed that as the atomic number of alkali-activated ions increased in the related geopolymer cement pastes, the geopolymerization process was drastically accelerated. The atomic number of alkali metal ions was proportional to the basicity of the system environment. The geopolymerization process was indicated to be drastically increased with the rising atomic number of alkali-activated ions in the corresponding geopolymer by comparing geopolymer cement pastes activated by various types of alkali metal ions. The atomic number of alkali metal ions directly determined the system environment’s basicity. Therefore, the geopolymerization reaction process of the corresponding geopolymer indicated that the reaction rate as well as the total geopolymerization degree had been enhanced simultaneously as the atomic number of alkali-activated ions increased.

Krishna et al. [43] focused on developing environmentally friendly geopolymer technology by studying the effectiveness of using acidic and alkaline activators as binding agents in the preparation of geopolymer composites. Acid-based activators such as phosphoric acid and aluminum-phosphate-based activators are potential alternatives to alkali-based activators. It has been reported that better mechanical and microstructural properties of geopolymer are obtained when produced by phosphate-based activators than when produced by alkali-based activators. The absence of alkali ions and an increase in bridging oxygen in the former geopolymer compared to the latter are the reason for the superior performance.

While most of the research has focused on NaOH and KOH, Suarez et al. [44] used lithium carbonate (Li_2_CO_3_), producing Li-geopolymer/β-Eucryptite with the aim to study the low-temperature reaction between kaolin and Li_2_CO_3_. The diffusion of lithium experienced an increase following the elimination of surface OH− and C contents from amorphous geopolymer particles during high-vacuum SPS treatments. This elimination process triggers the nucleation of nanocrystals, resulting in a uniform distribution throughout the material. Based on the findings, it can be concluded that the inclusion of lithium in the reactional medium facilitates the destabilization of the kaolin network. Choosing a suitable raw materials source is essential, as it is closely related to the properties obtained. A good property is required to achieve an excellent performance of the synthesized geopolymer. 

### 2.2. Properties of Geopolymer

In terms of strength, hardness, and chemical stability, geopolymer possesses characteristics that are essentially identical to those of regular cement. It offers exceptional resistance against acid and fire attacks. It provides a rapid setting time without reducing compressive strength, minimal creep, and low shrinkage. By exploiting environmentally harmful by-products, geopolymer also offers a lot of potential for creating green and environmentally favorable materials. Due to its many advantages, it can be used to create stabilized pavement that is environmentally beneficial [42].

Alkaline cements, also known as geopolymers, are a new family of aluminosilicate binders that have attracted extensive research because of their two key benefits: low energy usage and zero CO_2_ emissions during preparation [45]. Some aluminosilicate substances, such metakaolin or fly ash, produce a three-dimensional alkaline aluminosilicate hydrate when hydrated in the presence of alkalis. When kaolin, an industrial mineral, is heated between 650 and 900 °C, aluminosilicate solid raw material is produced. Kaolin’s solubility in alkaline media is increased by the thermal dehydroxylation process, making metakaolin an excellent raw material for the geopolymerization-based manufacture of inorganic polymers.

A considerable amount of the amorphous aluminosilicate phase, which is often readily dissolved in sodium hydroxide solutions, is typically present in metakaolin; hence, its solubility is expected [46]. The polycondensation process involves creating an alkali (Na or K) sialate–siloxo cross-linked polymeric network in the case of geopolymer concrete, which quickly hardens and acts as a bonding agent. These polymeric chains enable the fusion of solid granules to produce geopolymers, which are solid, compact materials. The most important properties are their ability to develop high mechanical strength [47] in a short period and at a moderate temperature (T < 100 °C) and their excellent durability. The research that has been done so far on their fire resistance and other high-temperature characteristics is intriguing. All findings concur that these materials outperform Portland cement at temperatures of 600 to 800 °C or higher; in most cases, their compressive strength rises after cooling. Most published literature states that these materials reach their critical point at temperatures between 600 °C and 800 °C. In that temperature range, these materials’ dimensional stability is severely compromised, their bending strength is significantly diminished, and their compressive strength increases. This result can be attributed to the partial sintering that takes place at these temperatures or a little bit higher. For geopolymers based on metakaolin, 900 °C is a crucial transition temperature because it causes localized partial melting and coagulation of the geopolymer. At this temperature, significant interspaces that separate the locally melted structures still exist.

### 2.3. Geopolymer Exposure to High Temperature

One of the distinctive characteristics of geopolymers is that they vary in bulk density and, therefore, in mechanical strength when subjected to high temperatures. In general, all geopolymers that were heated maintained their cubic shape up to 800 °C without being destroyed or changing in dimension. This was supported by Skvara et al. [48] for fly ash geopolymers. However, according to Badanoiu et al. [49], partial melting and softening occurred in the foamed geopolymers made of glass and red mud.

The bulk density of the unexposed geopolymers, which were held at room temperature, slightly decreased. This was attributable to the minimal moisture loss from evaporation during the room-temperature curing procedure. For unfoamed geopolymers, samples heated at 200 °C, 400 °C, and 600 °C showed the greatest drop in bulk density [50]. The release of water from the structure caused thermal shrinkage of geopolymer samples at high temperatures, which degraded the geopolymer structure. The mass loss of geopolymers occurred at temperatures lower at 800 °C. This could be a result of the samples’ expansion brought on by heat, or the geopolymer matrix’s densification, which makes up for the mass loss. This trend was supported by Duxson et al. [51], whereby the geopolymer sample shrunk at the beginning of temperature exposure and finally densified at a higher temperature. On the other hand, a larger mass loss was caused by the higher exposure temperature for foamed geopolymer. The variance in density reduction was caused by the differences in the water content of the geopolymers.

When heated from 200 °C to 800 °C, the strength of geopolymers decreased. This was in agreement with the fact that geopolymers’ bulk density dropped as temperature increased. After 400 °C, the strength deteriorated more slowly. As the temperature was raised, water vapor loss caused the structure to become more porous, which decreased its strength. Both Zhang et al. [52] and Zuda et al. [53] agreed that geopolymer strength decreased at high temperatures of up to 800 °C. In addition, based on the observation by Lemougna et al. [54], the dense volcanic ash geopolymers thermally treated between 250 °C and 900 °C showed a similar tendency toward weakening. A contrasting result was reported by Bakharev [55], wherein the compressive strength and average pore size of fly ash geopolymer tended to increase during heating. Fly ash geopolymers often showed greater strength deterioration. The thermal performance of fly ash geopolymer may be improved by adding metakaolin. Despite the lowest mass loss at 800 °C, the strength did not improve. The distortion of the geopolymer matrix and structure can account for this. The overall structure of the geopolymer was affected internally by the migration of water to the surface at high temperatures. The strength of geopolymers was also diminished concurrently by expansion in the geopolymer samples. The past research of geopolymer expose to high temperature are as summarized in Table 2 below.

The compressive strength of geopolymers degraded when exposed at 200 °C compared to unexposed geopolymer. When heated at 400 °C or 600 °C, the strength drastically decreased. In another way, geopolymer samples heated to 800 °C maintained greater strength. Zhang et al. [56] observed a different strength pattern, with the fly ash geopolymers maintaining their strength at 400 °C and showing a greater strength increase at 800 °C than the geopolymers that were not exposed to heat. The development of a greater Si–Al matrix was thought to be influenced by the comparatively higher SiO_2_ and Al_2_O_3_ compositions of fly ash, which was employed in the study. On the other hand, based on Bernal et al. [57], up to 800 °C, the strength of metakaolin geopolymer decreased, and after 1000 °C, it increased. Slag–metakaolin geopolymers tended to lose strength gradually between 200 °C and 1000 °C. It is clear that foamed geopolymer generally exhibited less strength depreciation than unfoamed geopolymer. It was expected that the pore structure of foam material would promote heat transport and reduce thermo-mechanical damage. The porosity made it possible to quickly remove water, which increased thermal resistance. This statement was further supported by Zhao and Sanjayan [58], who stated that the internal pore structure allowed the quick escape of water vapor that reduced the pore pressure. The performance of geopolymers as temperature increased was governed by the mass loss of the materials and the thermal deformation brought on by water evaporation. When the temperature increased, structural water or water in the pore cavities moved rapidly and evaporated through the surface [59].

In order to analyze the morphological features of the geopolymer, SEM micrograph analysis was conducted on both unexposed and exposed samples. Ramli et al. [60] conducted a study to investigate the impact of sintering temperature on the pore structure. The findings revealed that when the geopolymer was heated to temperatures of 900 °C and 1100 °C, a significant presence of large pores and cracks were observed, as in Figure 4 below. These interconnected pores contributed to an overall increase in the internal porosity.

In another study by Polat et al. [61], geopolymer foams were exposed to various temperatures, 600, 700, 725, and 750 °C, to observe the mechanical behavior of the geopolymer. It is reported that when the temperature was above 700 °C, the glass particles were semi-melted and diffused into each other, as can be observed in Figure 5. The research findings revealed that upon exposure to 700 °C, the glass in the sample began to partially melt. Furthermore, sintering above this temperature led to a decrease in foam density due to the decomposition of thermonatrite, which released CO_2_ gas.

A unique feature of geopolymer is that it can be converted into ceramics when exposed to a high temperature. Not only will it produce a glassy surface on ceramics, but it will also improve the characteristics of the sintered geopolymer. 

## 3. Geopolymer for Ceramic Application

Recycling has become a crucial strategy for tackling the environmental problem, because of the rapid advancement of technology and society, the depletion of natural resources, and the rise in solid waste. Thermal power plants produce a type of industrial solid waste known as coal fly ash, which is responsible for substantial environmental pollution, including haze and fog produced by random accumulation. In the meantime, glass-ceramics can be entirely manufactured using the oxides present in coal fly ash, such as silica, alumina, calcium oxide, and iron oxide.

Beginning in the 1980s, when Deguire started producing glass-ceramics using coal fly ash as a raw material, many studies have gradually emerged, focusing primarily on two techniques: one involves adding a nucleating agent to control crystallization from a parent glass, and the other involves sintering parent glass to increase crystallization. At a high temperature (about 1500 °C), the raw materials must be melted in order to create the desired glass-ceramics. This process consumes a lot of energy. Researchers suggested a method of directly sintering to create glass-ceramics utilizing relatively low processing temperatures in order to overcome the drawbacks of the preceding two technologies’ excessive energy consumption. The ceramic sintering procedure is the same as the direct sintering technique. Direct sintering has the ability to minimize processing temperature and energy consumption compared to conventional glass-ceramic preparation processes. It can also utilize ceramic manufacturing equipment for mass production, which lowers production costs. Although direct sintering provides several benefits, the products’ qualities (such as water absorption and bending strength) are marginally inferior to those produced using conventional processes. Nevertheless, according to some studies, adding glass powder to fly ash can enhance the performance of glass-ceramics. From the standpoint of fly ash resource utilization, it is imperative to utilize fly ash as much as possible [62].

The surface energy of the powder, which is directly connected to the powder’s particle size, acts as the driving force throughout the sintering process. When manufacturing glass-ceramics using conventional procedures, it has been discussed how the parent glass’s grain size affects crystallization and sintering. The preparation of glass-ceramics with fine parent glass particles implies a low sintering temperature, high strength, and a smooth surface [62]. Many researchers have performed research to improve the performance of geopolymer ceramics, such as Lu et al. [63], who investigated the size influence of slag powders on the properties of glass-ceramics through direct sintering, whereas Erol et al. [64] discovered that the characteristics of the sintered material made from coal fly ash were influenced by the raw material’s particle size in addition to the sintering temperature and time.

### 3.1. Geopolymer Ceramics

Geopolymer ceramics provide advantages in compressive strength, low permeability, good chemical resistance, and excellent fire resistance behavior [65]. Due to its promising properties, further research needs to be performed to understand the science behind geopolymer technology. Since sintering temperature plays a significant role in determining the properties of geopolymer ceramics, it is imperative to conduct a thorough investigation into the effects of heat application during the sintering process on the ceramic body.

In a study, Villaquirán-Caicedo and de Gutiérrez [66] produced metakaolin-based geopolymer using KOH and eco-friendly silica sources, rice husk ash and silica fume. In their finding, structural densification could be observed as the sintering temperature increased from 25 °C to 1200 °C. This was driven by the dehydration of the produced geopolymeric products, and the structural changes generated. At a temperature of 900 °C, heat exposure caused microcracks growth, as the capillaries contracted during the dehydration and dehydroxylation of the geopolymeric gel. Densification happened at 1200 °C, forming a crystalline structure, leucite. At the optimum sintering temperature, 1200 °C, compressive strength was at its highest due to the densification and crystallization of the geopolymer gel. 

Jaya et al. [67] studied the effect of temperature on the properties of nepheline ceramics. It was observed that sintering the ceramic up to 1200 °C reduced the size of the particles, thus reducing the total surface area. This phenomenon was related to water removal during the sintering process and the development of growth in the crystalline phase. Heating up to 1200 °C assisted in strengthening the bonds between the particles’ constituent parts and prevented thermal degradation at high temperatures [68,69]. Amorphous geopolymer transforms into crystalline nepheline ceramic when heated to high temperatures. Higher temperatures increase consolidation and make for a more uniform microstructure, according to the microstructural analysis of samples with the highest strength.

He and Jia [70], in their study on producing low-temperature sintered pollucite from Cs-based geopolymer using synthetic metakaolin, concluded that the ceramics showed a low sintering temperature range which initiated at 800 °C, ended at 1200 °C, and indicated complete crystallization after being heated to 1200 °C. The appearance of the pollucite at a low temperature increased the viscosity of the material at a high temperature. Thus, the sintering stage was postponed to a higher temperature range.

In research conducted to study the performance of blended-ash geopolymer concrete at elevated temperatures by Hussin et al. [71], in order to assess the mass loss, strength, and microstructural changes brought on by thermal impact, the samples were heated up to 800 °C. It was discovered that exposing the concrete to high temperatures boosted its strength. The reason for this rise in compressive strength was that the geopolymer melted at a higher temperature due to the low Na^+^ diffusion coefficient at high temperatures. Meanwhile, the study by Hsieh et al. [72] concluded that the structure became denser as the temperature rose, which was related to the potassium found in the alkali activator being dehydrated.

As sintering ceramic requires a high temperature, a study by Zawrah et al. [73] used nano sand to improve the properties of kaolin-based geopolymer while studying the low-rate sintering of the prepared geopolymer. Nepheline was generated in greater quantities as the sintering temperature was raised. The green geopolymer’s porosity essentially rose at around 800 °C because of water evaporation, recrystallization, and sintering. At 1000 °C, the porosity of the sintered samples decreased, and bulk density increased as there were more liquid phases formed and there was greater grain diffusion, which increased the strength of the ceramics.

### 3.2. Properties of Geopolymer Ceramics

Geopolymer is amorphous, whereas the aluminosilicate materials used harden at ambient temperature, then are converted into ceramic when heated at a high temperature. The sintering process not only affects the evolution of the phase composition but also influences the microstructure and interface conditions of the geopolymer [74]. Leucite, a geopolymer ceramic, has been produced for a variety of uses, including dental porcelain, refractories, and structural ceramic materials. Leucite is frequently used in these applications due to its high thermal expansion coefficient and comparatively high melting point of 1693 °C. As a low-cost heat-resistant structural material, for non-flammable heat-resistant components, etc., geopolymer ceramic can also be employed [4].

Ceramics made of high-purity geopolymer can be synthesized using metakaolin-based geopolymer precursors. Additionally, by adjusting the alkali or silica concentration, the thermal expansion of the ceramic can be customized. For instance, K exhibits a higher thermal expansion than Cs, which exhibits a lower thermal expansion. Geopolymer will convert into crystalline phases after being heat treated, and these phases offer good mechanical and thermal properties. The development of a composite matrix form is strengthened because of heat treatment, which is visible in the mechanical properties of geopolymer ceramic following treatment at 1100 °C. 

Mechanical testing was used to measure the flexural strength, hardness, and toughness of pressed fired discs. Using SEM and TEM analysis, the relationship between the microstructure and mechanical characteristics was examined [75]. In the current study, a structural grid made of ceramic struts and void space porosity was visible in the porous structure when it was visualized. Increasing the sintering temperature will reduce the porosity of the ceramic formed, resulting in a denser microstructure, thus improving the hardness of the geopolymer ceramic. As a consequence, the mechanical properties could be enhanced [76]. Several researchers have undertaken further studies on the sintering process based on several factors that can influence the process (e.g., the sintering profile, heating rate) to improve more on the performance of the ceramics formed. 

## 4. Sintering Mechanism of Ceramic Materials

Ceramics’ chemical and micro- and crystallographic compositions determine how well they perform [77,78,79]. The morphology and crystallographic structures need to be precisely controlled in order to produce desirable microstructures made up of grains with the right chemical composition; in turn, processing principles and appropriate sintering techniques depend on the working of kinetic mechanisms. In the development of nano-ceramics and multifunctional ceramic composites, the precise manipulation of kinetics is a significant problem, because the “kinetic windows” for retaining small grain structures and/or for preventing undesirable reactions from occurring are typically rather tight [80].

Traditionally, the production of ceramics has involved the sintering of powders or powder compacts at high temperatures for prolonged periods of time. This procedure results in the removal of pores and coarsening of the particles via mass transport by grain boundary, surface, and lattice diffusion, demonstrating the involvement of at least two connected processes, namely densification and grain coarsening/growth [81,82]. Micron-size precursor powders have been the subject of numerous sintering studies in the past, and it was discovered that surface diffusion is only effective during the initial stages of sintering when it comes to coarsening of the particles. Surface diffusion is ineffective when it comes to densification, for which grain boundary diffusion has been cited as the primary mechanism. The fact that the final stage of sintering is consistently associated with rapid grain growth [83] is frequently observed evidence that the formation of grains is also aided by grain boundary diffusion, which is more active at higher temperatures than surface diffusion. Surface diffusion plays an important role in the densification process, since the usage of nano-sized precursor powders suggests the presence of high surface areas that in turn affect the development of the particle/pore structure during sintering.

Solid-state sintering of crystalline materials (SSS), solid-state sintering of amorphous materials (sometimes known as “viscous sintering”), and liquid-phase sintering (LPS) of crystalline materials are the three different types of sintering [84]. The bonding of particles and the densification of powder compacts occur regardless of the sintering technique. SSS and LPS both see an increase (coarsening) of grains (particles). Grain growth, albeit minor at first, especially in SSS, becomes significant with densification and has a huge effect on the ultimate density and the microstructure that comes out of it. Amorphous materials undergo densification due to the viscous flow of substances in which there is no physical barrier between the particles [85,86].

The connection between the solid and the porous phase has led to the conventional division of sintering into three overlapping stages: initial, intermediate, and final. Adjacent particle bonding, substantial neck expansion, and restricted densification are the characteristics of the early stage [87]. A connection exists between the solid and porous phases. The compact powder goes through a significant amount of densification in the intermediate stage, and at this point, the solid and porous phases are joined. The solid phase is linked throughout the last phase, but the pores are isolated. This stage of grain growth for crystalline materials is when pores and grain boundaries interact to influence the evolution of the microstructure [88].

Both in SSS and LPS, the sintering of crystalline powder compacts involves the transfer of materials from an atom source (or sources) to an atom sink (or sinks) through the separation (an interface reaction), movement (primarily through diffusion), and attachment (an interface reaction) of the source’s atoms. The surface of a small grain grows into the surface of a large grain across the grain boundary (for SSS) or via a liquid phase in a manner similar to this (for LPS). Therefore, the slower process, such as diffusion or an interface reaction, which is a feature of serial processes, must control the kinetics of bonding, densification, and grain coarsening [89]. Conventionally, however, the analysis and predictions of densification and grain formation in crystalline materials have been based on the presumption that diffusion controls their kinetics. Recent research has revealed that this presumption is only true for crystalline systems with rough (atomically disordered) surfaces.

Sintering factors such as heating rate [90,91], sintering temperature [92], and holding time [93] have a crucial effect on the final density and mechanical properties of a sintered sample. Several studies had been done to aid in a further understanding of the sintering mechanisms of ceramic materials. Table 3, below, summarizes the sintering profiles of the sintering ceramics processes.

Shen et al. [94] studied the effects of debonding and sintering profiles on transparent ceramics. Three sintering schemes were prepared at the same sintering temperature of 1800 °C with different sintering rates (5 and 10 °C/min) and holding times (5 and 10 h). The outcome demonstrates that when the sintering heating rate was lower, the transmittance was relatively high. The green bodies were slowly and evenly heated, reducing interior bodily tension. A lower heating rate facilitated the crystallization process in geopolymer ceramics by enabling a gradual and controlled transformation. The diffusion of the amorphous phase occured during this process, ultimately leading to the formation of the final crystalline structure of the ceramics. By heating the material slowly, it became possible to control the growth rate of the crystals, achieve a desired grain size, and eliminate any existing pores.

Meanwhile, Frueh et al. [95] studied the variables affecting the accuracy of the master sintering curve, which was used to examine the entire sintering profile of ceramic powders. The result shows high heating rates of 35–150 °C/min contributed to a lower Q value of 290 kJ/mol. Meanwhile, a Q value of up to 1064 kJ/mol was reported by Shao et al. [96] for granulated and dry-pressed alumina powders. The higher Q values were attributed to the densification impact of slower heating rates (0.5 and 5 °C/min). Due to the study of relative duration, the material was heated under conditions that were favorable for surface and grain boundary diffusion, with the heating rate affecting densification. It was determined that surface diffusion and particle coarsening were encouraged by the slow heating process. Before reaching the temperatures where crystal densification took place, samples heated slowly spent more time at lower temperatures and underwent greater particle coarsening.

In another study by Zhang et al. [97], an addition of foaming agent resulted in the formation of phase boundaries between the glass phase and nucleation sites. This, in turn, promoted a nucleation and crystallization process. Meanwhile, Pei et al. [98] investigated the effect of an addition of CaF_2_ as a fluxing agent on the sintering and crystallization of Cao–MgO–Al_2_O_3_–SiO_2_ glass-ceramics. The presence of exothermic peaks indicated the occurrence of crystallization during the heating process. With a higher heating rate, the glass transition and crystallization peaks of the sample shifted towards higher temperatures, which can be attributed to the hysteric nature of the heat effect.

The effects of heating rate, sintering temperature, and holding time on the process of densification, the microstructure, and the physical characteristics of sintered samples was studied by Li et al. [99]. In addition, the microstructure of a sintered sample was found to be significantly altered by the quantity of graphite foil layers used to pack the powder, which was studied by Moshtaghioun et al. [100]. 

### 4.1. Method of Sintering Kinetics

Researchers such as Liu et al. [101], Ptáček et al. [102], and Fan et al. [103] have studied the kinetics of sintering to observe the sintering behaviors of the ceramic, as kinetic study provides evidence for the mechanisms of the sintering process. By the time of their research, the knowledge of reaction mechanisms was of practical use in deciding the most effective way to cause a reaction to occur, thus making it possible to choose reaction conditions that favored one path over others. 

In Table 4, below, are compiled the types of kinetic models used to study sintering kinetics.

In a study by Liu et al. [101], XRD, non-isothermal DTA, time-resolved energy-dispersive powder diffraction, and DSC were all utilized to explore mullitization kinetics using kinetic models with particular experimental expressions. The bimodal Johnson–Mehl–Avrami (JMA) model was used to determine nucleation growth. The MakiPirtti–Meng equation was supported by the isothermal shrinkage data, demonstrating its applicability to the sintering process. Three phases made up the shrinking process. Activation energy (E_a_) was attributed to a wide range of phenomena in the initial phases (950–1100 °C), including the diffusion of the glassy phase and grain boundaries, the transformation of alumina and silica, and the production of spinel. E_a_ was assigned the primary accountability for the second-stage (1200–1300 °C) mullitization reaction in spinel and glassy silica. In the third stage (1300–1450 °C), glassy silica diffused and interacted with corundum to make crystalline orthorhombic mullite, and some enormous pores were created by small pores combining, diffusing, and vanishing. This process of pore formation is what dominates Ea.

Ptacek et al. [102] used Kissinger’s and Eyring’s laws to study the mechanisms and thermodynamics of the sintering process. The heating rate had a big impact on the constant-rate-of-heating (CRH) sintering behavior of compacted kaolin powder as well as on the kinetics and mechanisms of the sintering processes. It has been determined that faster heating produces ceramics with less cristobalite. The crucial kinetic factors for mullite and cristobalite synthesis can be distinguished with a higher heating rate.

Meanwhile, Fan et al. [103] modified the Kissinger equation to calculate the crystallization dynamic. The activation energy increased with the decrease in the liquid–solid ratio. The outcome shows that basicity had an impact on the crystallization mechanism. The decrease in modifier oxides and the rise in glass-forming oxides support this conclusion.

### 4.2. Model of Sintering

Toussaint and De Wilde [104] studied the sintering model for latexes in 1997. A new definition was derived regarding the minimum film formation temperature. The model also makes it possible to calculate the mechanical properties that a polymer has to have in order to successfully sinter under specific drying conditions.

In 2001, Oscar Prado et al. [105] proposed three sintering stages—a pure “Frenkel” (F) first step, a mixed “Frenkel/Mackenzie–Shuttleworth” stage, and a final, pure “Mackenzie–Shuttleworth” (MS) phase—which were presented as a model to describe the sintering kinetics of polydisperse glass particles. According to the model, sample shrinkage is taken into account as the total of the partial shrinking of a number of clusters, each of which is composed of particles of identical size and exhibits independent F or MS behavior. The whole set of clusters closely resembles the actual particle size distribution of the specimen. It is now understood to be possible for particles of various sizes to form necks due to the concept of neck-forming ability, ξ_r_, which loosens the clustering requirement. According to reports, the model offers a tool for calculating the sintering kinetics of actual glass powders at any temperature and size distribution, hence reducing the need for laboratory tests.

In a study by Prado et al. in 2005 [106], different models were developed to account for the kinetics of sintering. The shrinkage rates of two equal particles with nearly spherical centers could be calculated using the Frenkel (F) model of sintering, which explains the early phases of the sintering of spherical and monodispersed vitreous particles. When the surface area is reduced, less energy is produced, and viscous flow, which is in charge of mass transfer and densification, uses that energy instead. Within the first 10% of linear shrinkage, the Frenkel model is generally accurate. 10% of linear shrinkage causes compacts with a relative “green” density of 0.6 to become compacts with a relative density of 0.8.

A matrix with spherical monodispersed pores underwent the final stages of sintering according to a model created by Mackenzie and Shuttleworth (MS). In the model’s final stages of densification, when energy dissipation is connected to the change in surface area, the rate of densification of the viscous body with closed spherical pores is explained [107]. Higher relative densities above 0.9 are covered by this model. Thus, none of the aforementioned models can be used to fill the space between densities 0.8 and 0.9. When the compact’s initial density is 0.15 or less, as occurs in materials made of gel, this gap is significantly bigger. Scherer examined this issue and took into account a geometric array of sintering particles that resembled the structure of dry gels. Similar to Frenkel, Scherer made the assumption that the energy required for viscous flow was equal to the energy change brought on by the surface area reduction. His description of the sintering of compacts from a very low relative density was successful. The results of this model are essentially unaffected by geometrical characteristics because it may be used to densify bodies with high green densities (even for particle arrangements that differ from those of dry gels). The results of the Scherer and Frenkel models, for instance, virtually exactly match the first 10% of linear shrinkage. When sintering pure SiO_2_ preforms made by flame-hydrolyzing SiCl_4_ with a pore size dispersion, the Scherer model was successfully used.

The F and MS regimes may then coexist with to the cluster model, which was further developed to describe the sintering of compacts with any particle size distribution (for clusters of different particle sizes). According to this concept, the sample shrinkage is regarded as the total of the partial shrinkages of a number of clusters, each of which is composed of particles of identical size and exhibits a distinct F or MS behavior. The overall set of clusters closely resembles the specimen’s actual particle size distribution.

Based on these concepts, there are three stages of sintering in a compact: the first is pure “Frenkel,” the second is mixed “Frenkel/Mackenzie–Shuttleworth,” and the third is pure “Mackenzie–Shuttleworth”. In order to account for the creation of necks between particles of various sizes, the notion of “neck-forming ability—ξr “ was introduced, modifying the assumption of clustering.

### 4.3. Effects of Sintering and Properties of Common Materials

Ban and Choi [108] studied the effect of sintering on the conductivity of lithium–lanthanum titanates’ grain boundaries. The Li content dropped as the sintering temperature rose from 1100 °C to 1350 °C due to Li evaporating during sintering. Following sintering at 1200 °C or more, the samples were virtually entirely dense. Due to the expanding grain size, the grain boundary conductivity increased quickly with sintering temperature [109,110]. The change in grain boundary conductivity was primarily explained by the microstructural evolution when the grain conductivity was nearly independent of the sintering condition. Very few pores were found after sintering at temperatures exceeding 1200 °C. As the sintering temperature increased, the samples’ grain size increased continuously [111]. Between ambient temperature and 300 °C, the samples’ impedance was measured, and from the impedance patterns, the conductivity of the grains and their boundaries were determined. The Li content decreased as the sintering temperature increased from 1100 °C to 1350 °C, with a 50 °C step, because Li evaporated during sintering. The high sintering temperature, including some other factors such as holding time, affecting the grain size growth which responsible to the properties of the end ceramics [82].

Ayyappadas et al. [112] performed a study wherein the composite was exposed to a high temperature of 900 °C with a holding time of 60 min and a heating rate of 5 °C/min (conventional) and 20 °C/min (microwave). The processing cycle time was reduced by 63% when compared to the conventional procedure due to the fact that all the composites were found to couple with the microwave field well. The homogenous distribution of graphene in the copper matrix was discovered by microstructural investigation. Due to graphene’s superior lubrication and increased hardness, copper–graphene composites showed exceptional wear resistance. Porosity was found to significantly affect the electrical conductivity values.

Asadikiya et al. [113] investigated the relationships between the density and hardness of sintered spark plasma and the impacts of sintering temperature, heating rate, and holding time. It was concluded that when all other sintering parameters are constant, a higher sintering temperature can produce a sample that is denser in a fixed phase region [114,115]. However, it is important to tune the sintering temperature to have the least impact on irregular grain formation, which is very likely at higher temperatures.

There have been many attempts by researchers to improve the characteristics and properties of geopolymer ceramics, mostly by studying the impacts of raw materials such as metakaolin, kaolin, rice husk, and fly ash, and their sintering profiles, during the fabrication of ceramics. To conclude, most researchers have proven that sintering geopolymers at optimum temperatures and under optimum conditions do enhance the chemical and mechanical properties of these ceramics.

## 5. Summary and Future Work

From the review that has been done, it can be concluded that the sintering mechanism contributes a significant effect to the properties of the produced ceramics. Commonly, a high temperature was necessary to sinter the ceramics. However, by using geopolymer materials, this process required a low cost and low energy consumption but could produce excellent mechanical and physical properties. Furthermore, the geopolymerization method used can also reduce greenhouse emissions, thus lowering air pollution. Geopolymer converts into crystalline phases with outstanding mechanical and thermal properties after being heat treated. The final result, geopolymer ceramic, is a non-flammable heat-resistant component and a low-cost heat-resistant structural material.

This review focused on the effect of the sintering mechanism on the crystallization kinetics of geopolymer ceramics. Based on past research, it can be concluded that increasing the sintering temperature will reduce the porosity of the ceramic formed, resulting in a denser microstructure, thus improving the hardness of the geopolymer ceramic. The mechanical properties are then enhanced. Lowering the heating rate will increase the transmittance, as it resulted in the green bodies being heated evenly and slowly, thus controlling the growth rate and grain size of crystals and eliminating pores. Furthermore, a slow heating rate is advantageous because it allows for surface and grain boundary diffusion to occur for a longer period while the material is heated. Before reaching the temperatures where crystal densification takes place, samples heated slowly spend more time at lower temperatures and undergo greater particle coarsening.

However, there is still room for improvement in the functionality of geopolymer as a basic source for ceramics and in the sintering process of ceramics, as it is an important approach to manufacturing ceramics of controlled density and microstructure. The reaction mechanism of the complex reaction can be understood by conducting extensive research on the sintering kinetics of ceramic production. Numerous technologically significant materials and systems are currently being studied, and there are numerous essential lessons from the science of sintering that have been and will continue to be utilized.

## Figures and Tables

**Figure 1 materials-16-04103-f001:**
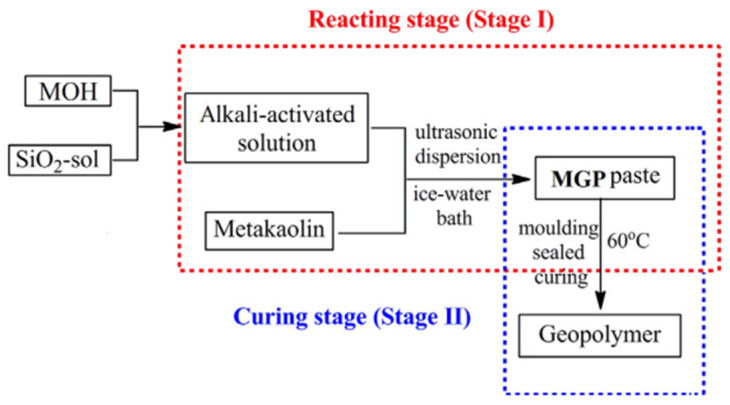
Diagram of a division of the geopolymerization process [27].

**Figure 2 materials-16-04103-f002:**
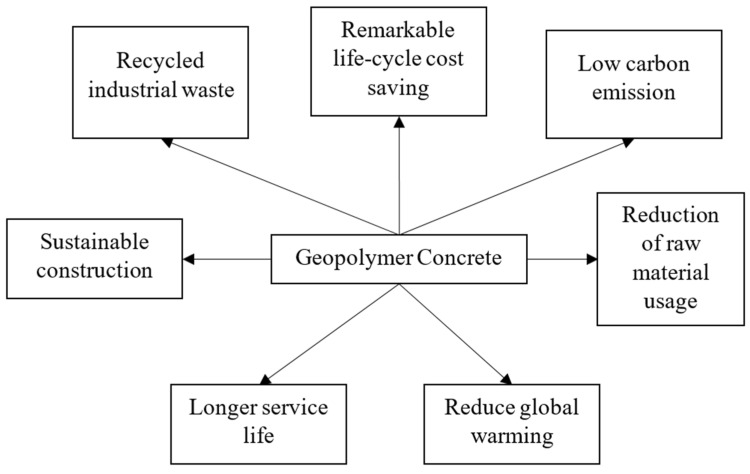
The application of geopolymer concrete in construction.

**Figure 3 materials-16-04103-f003:**
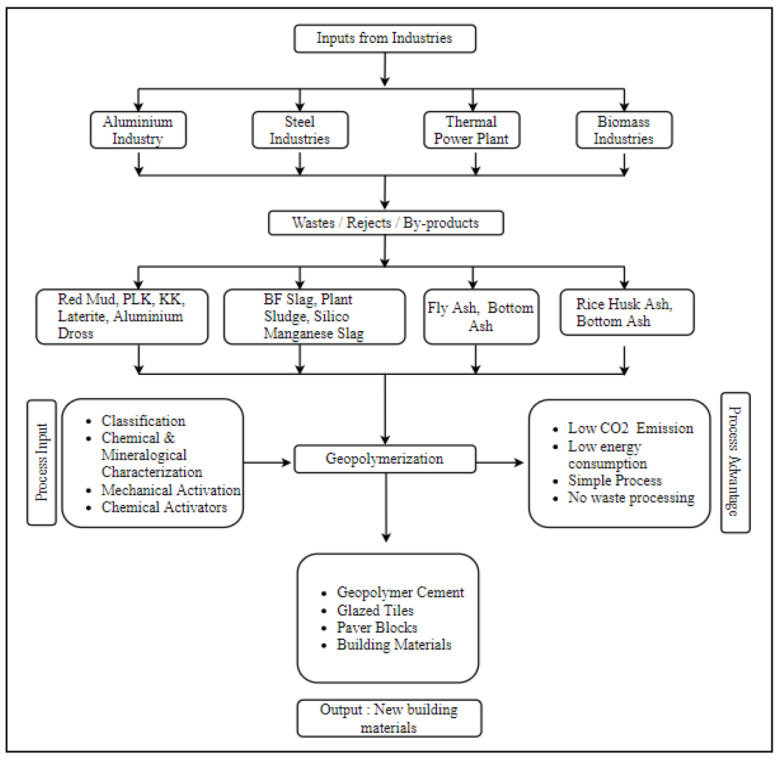
Process diagram of input sources from industrial waste to geopolymer synthesis [41].

**Figure 4 materials-16-04103-f004:**
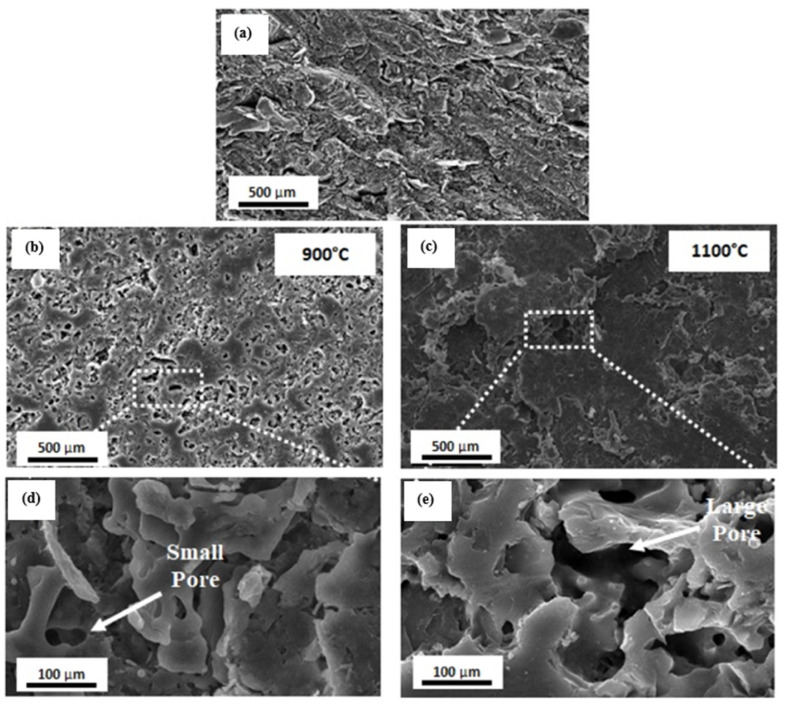
SEM micrograph of (**a**) unsintered, (**b**,**d**) sintered at 900 °C, and (**c**,**e**) sintered at 1100 °C kaolin-based geopolymer [60].

**Figure 5 materials-16-04103-f005:**
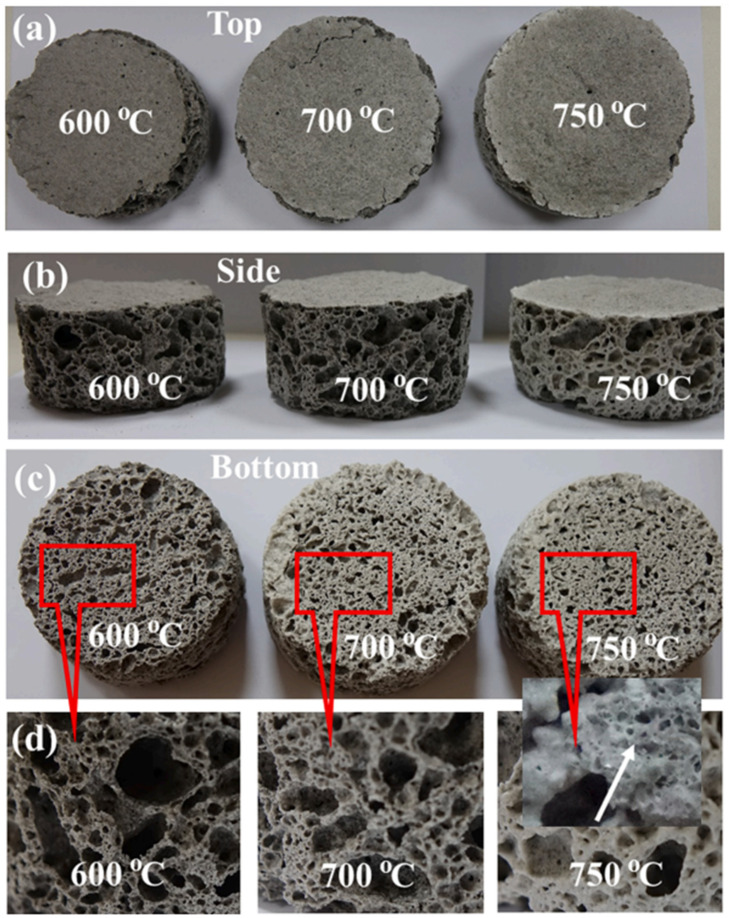
The images of sintered foams; (**a**) top, (**b**) side, and (**c**) bottom views, and (**d**) magnified images showing the cell structure at different temperatures [61].

**Table 1 materials-16-04103-t001:** Summary of past research on raw materials used and the properties.

Author	Raw Materials	Properties
Bong et al. [38]	Wollastonite	Improve the flexural strength Increase the compressive strength
Ren et al. [39]	Metakaolin with reinforcement of wollastonite, tremolite, short basalt	Increase the compressive strength Enhance the resistance to sulfate and chloride attack
Archez et al. [40]	Wollastonite	Improve the viscosity and mechanical properties

**Table 2 materials-16-04103-t002:** Summary of past research of geopolymer expose to high temperature and its findings.

Author	Exposed Temperature	Findings
Zhang et al. [56]	Up to 800 °C	The strength of fly-ash-based geopolymer increased as the temperature increased to 800 °C
Bernal et al. [57]	Up to 1000 °C	The strength of metakaolin-based geopolymer increased when further heated up to 1000 °C
Zhao and Sanjayan [58]	Up to 800 °C	Sintering of fly ash geopolymer increased the strength at 800 °C

**Table 3 materials-16-04103-t003:** Summary of past research on sintering profiles.

Author	Sintering Profile	Findings
Shen et al. [94]	Sintering temperature: 1800 °C Heating rates: 5 and 10 °C/min Holding time: 5 h, 10 h	The transmittance was high when the heating rate of sintering was low
Frueh et al. [95]	Heating rates: 35–150 °C/min	High heating rate attributed to the lower Q value More particle coarsening when heated during slow heating process
Shao et al. [96]	Heating rates: 0.5 and 5 °C/min	Heating rate influences the densification; slow heating rate is favorable

**Table 4 materials-16-04103-t004:** Types of kinetic models used to study sintering kinetics.

Researcher	Type of Model Studied	Founding
Liu et al. (2002) [101]	Nucleation growth by Avrami Johnson and Mehl model Bimodal Johnson–Mehl–Avrami (JMA) model	The shrinkage process is divided into three stages: First stage (950–1100 °C) Second stage (1200–1300 °C) Third stage (1300–1450 °C)
Ptáček et al. (2012) [102]	Kissinger’s and Eyring’s laws	Heating rate affecting the sintering behavior, kinetics and mechanism of the process Increasing the heating rate produced ceramic with low content of cristobalite
Fan et al. (2018) [103]	Modified Kissinger’s equation	Activation energy increased with the decrease in the liquid–solid ratio The mechanism of crystallization was affected by the basicity

## Data Availability

Not applicable.

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
