# Peer review of "Effect of Sintering Mechanism towards Crystallization of Geopolymer Ceramic—A Review"

_materials, 2023, doi:10.3390/ma16114103_

Round 1

Reviewer 1 Report

The manuscript gives an interesting review about geopolymers, their production, properties and application. It is well written and interesting also for a non-specialist. Its language is concise and fluid. There is hardly any criticism or suggestion, yet

-          The authors may prepare a schematic figure similar to Fig. 1, which should review all geopolymers mentioned, their derivatives and related materials, which special notes where thermal treatment is required.

-          For curiosity, the authors may mention in the introduction, that according to some studies (or speculations), the Egyptian Great Pyramid was also made of geopolymers

(Joseph Davidovits – several references in Wikipedia; Guy Demortier:  Are carbon micro-clusters in the Khufu pyramid blocks of organic origin? PIXE and PIGE reveal the construction of Giza's pyramids, Spectroscopy Europe 33 (5): 21–29).

There are further some small errors or inconsistencies:

Lines 229, 230: word order: (K2SiO3) shall follow immediately after ‘potassium silicate’.

Line 235: I assume ‘pozzolanic’ is an adjective, so some definite word is missing.

Line 417: sintering able – sintering is able (?)

Lines 431, 432: Lu, Erol – add references.

Line 447: capillary contract – capillaries contract

Line 505: A structural – a structural

Line 621: Glassy – glassy

Author Response

Dear editor,

We express our gratitude to both you and the reviewers for dedicating your valuable time to review our paper and share your valuable insights. Your comments have played a crucial role in enhancing the current version of the manuscript. The authors have diligently examined each comment and made every effort to address them comprehensively. We aim to meet your high standards through careful revisions of the manuscript. If you have any further constructive comments, we would be delighted to receive them.

Please find our detailed responses to each point below, with all modifications highlighted in yellow.

Thank you.

Regards,

Nur Bahijah Mustapa, MSc. Student,

Faculty of Mechanical Engineering Technology,

University Malaysia Perlis (UniMAP).

Reviewer 2 Report

The idea that this review wants to convey may be of interest for the application of geopolymers in different fields, but its development and content is incomplete and partial.

In order for this work to be published, the following points should be addressed:

1.- The authors mention “One of the most widespread source materials containing aluminosilicate in geopolymer concrete (GPC) is metakaolin (MK)” but do not give significant data on the reason for its use. They should provide more data on MK, for example, its particular structure formed by AlO5, which induces greater reactivity ( J.Sanz, et al. Aluminum-27 and Silicon-29 Magic-Angle Spinning Nuclear Magnetic Resonance Study of the Kaolinite-Mullite Transformation. Journal of the American Ceramic Society, 71(10), C418–C421. 1988).

2.- The case of another alkaline such as Li is not mentioned either, MK + CO3Li2 gives rise to a Li geopolymer (Suárez, M., Fernández, A., Díaz, L. A., Sobrados, I., Sanz, J., Borrell, A., Palomares, F. J., Torrecillas, R., & Moya, J. S. (2020). Synthesis and sintering at low temperature of a new nanostructured betaEucryptite dense compact by spark plasma sintering. Ceramics International, 46(11) , 18469– 18477. ) .

In the case of kaolinite activated with Na2CO3, it produced zeolite precursors but in the case of Li2CO3 activation β-Eucryptite precursors are formed.

-             3.-  In the section on Geopolymer Exposure to High Temperature, the authors mention for example “The overall structure of the geopolymer was affected internally by the migration of water to the surface at high temperatures. The strength of geopolymers was also diminished concurrently by expansion in the geopolymer samples”, clearly required to be given , in this as well as in other possible cases , data on the kinetics of the sintering process (sintering curves) and the effect on the microstructure with microgrphs  of the microstructures obtained before and after the sintering process , in the corresponding figures. Data of the mechanical resistances reached in Tables  with more explicit explanatory comments.

Figures explaining the geopolymer sintering process are completely absent in the MS. Without these figures and the relevant comments the MS cannot be published in Materials.

Author Response

(The authors gave the same response as above.)

Reviewer 3 Report

Review attached

Author Response

(The authors gave the same response as above.)

Round 2

Reviewer 2 Report

 The corrections on the MS are appropriate then I consider that the paper can be published in Materials.

Author Response

[Response to Reviewer 2]

Thank you for your prior comments, which helped us improve this manuscript. Your thoughtful and perceptive remarks provided the foundation for potential improvements in the current version.

Thank you once again.

Nur Bahijah Mustapa

Reviewer 3 Report

After reviewing the responses from the authors of the manuscript, the reviewer acknowledges the responses and comments. The manuscript may be published in Materials with minor revisions:

1) on page 11, line 417 is CO2, and it should be CO2

2) on page 12 line 480, the reference should be corrected

Author Response

[Response to Reviewer 3]

Thank you very much for your previous comments that helped us in improving this manuscript. It was your valuable and insightful comments that led to possible improvement in the current version.

Minor revisions:

  • CO2 has been revised to CO[page 11]
  • The reference on has been corrected [page 12]

We hope these revisions meet with your approval.

Thank you once again.

Nur Bahijah Mustapa